# Addressing Failure Prediction
# by Learning Model Confidence

**Charles Corbière[1,2]**
charles.corbiere@valeo.com

**Nicolas Thome[1]**
nicolas.thome@cnam.fr

**Avner Bar-Hen[1]**
avner@cnam.fr

**Matthieu Cord[2,3]**
matthieu.cord@lip6.fr

**Patrick Pérez[2]**
patrick.perez@valeo.com

[1]CEDRIC, Conservatoire National des Arts et Métiers, Paris, France
[2]valeo.ai, Paris, France
[3]Sorbonne University, Paris, France

## Abstract

Assessing reliably the confidence of a deep neural network and predicting its failures is of primary importance for the practical deployment of these models. In this paper, we propose a new target criterion for model confidence, corresponding to the *True Class Probability* (TCP). We show how using the TCP is more suited than relying on the classic *Maximum Class Probability* (MCP). We provide in addition theoretical guarantees for TCP in the context of failure prediction. Since the true class is by essence unknown at test time, we propose to learn TCP criterion on the training set, introducing a specific learning scheme adapted to this context. Extensive experiments are conducted for validating the relevance of the proposed approach. We study various network architectures, small and large scale datasets for image classification and semantic segmentation. We show that our approach consistently outperforms several strong methods, from MCP to Bayesian uncertainty, as well as recent approaches specifically designed for failure prediction.

## 1 Introduction

Deep neural networks have seen a wide adoption, driven by their impressive performance in various tasks including image classification [25], object recognition [43, 33, 37], natural language processing [34, 35], and speech recognition [18, 15]. Despite their growing success, safety remains a great concern when it comes to implement these models in real-world conditions [1, 19]. Estimating when a model makes an error is even more crucial in applications where failing carries serious repercussions, such as in autonomous driving, medical diagnosis or nuclear power plant monitoring [32].

This paper addresses the challenge of failure prediction with deep neural networks [17, 20, 16]. The objective is to provide confidence measures for model's predictions that are reliable and whose ranking among samples enables to distinguish correct from incorrect predictions. Equipped with such a confidence measure, a system could decide to stick to the prediction or, on the contrary, to hand over to a human or a back-up system with, *e.g.* other sensors, or simply to trigger an alarm.

In the context of classification, a widely used baseline for confidence estimation with neural networks is to take the value of the predicted class' probability, namely the *Maximum Class Probability* (MCP), given by the softmax layer output. Although recent evaluations of MCP for failure prediction with

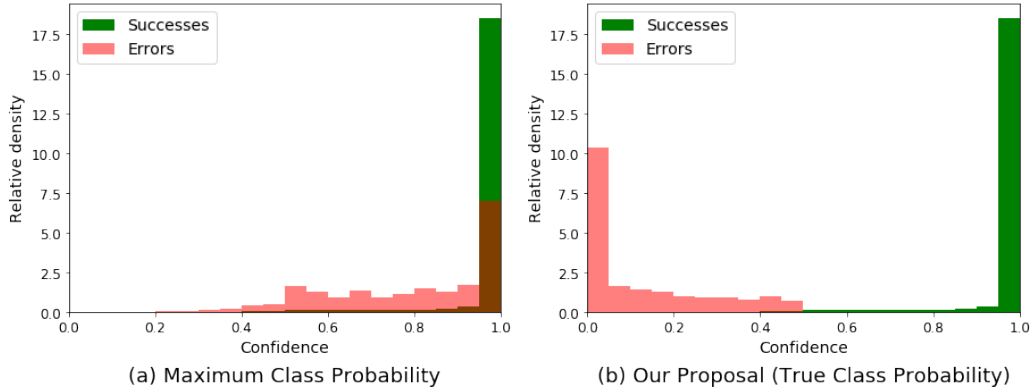

Figure 1: When ranking test samples according to *Maximum Class Probability* (a), output by a convolutional model trained on CIFAR-10 dataset, we observe that correct predictions (in green) and incorrect ones (in red) overlap considerably, making it difficult to distinguish them. On the other hand, ranking samples according to *True Class Probability* (b) alleviates this issue and allows a better separation for failure prediction. (Distributions of both correct and incorrect samples are plotted in relative density for visualization purpose).

modern deep models reveal reasonable performances [17], they still suffer from several conceptual drawbacks. Softmax probabilities are indeed known to be non-calibrated [13, 40], sensitive to adversarial attacks [12, 44], and inadequate for detecting in- from out-of-distribution examples [17, 30, 26].

Another important issue related to MCP, which we specifically address in this work, relates to ranking of confidence scores: this ranking is unreliable for the task of failure prediction [41, 20]. As illustrated in Figure 1(a) for a small convolutional network trained on CIFAR-10 dataset, MCP confidence values for erroneous and correct predictions overlap. It is worth mentioning that this problem comes from the fact that MCP leads by design to high confidence values, even for erroneous ones, since the largest softmax output is used. On the other hand, the probability of the model with respect to the true class naturally reflects a better behaved model confidence, as illustrated in Figure 1(b). This leads to errors' confidence distributions shifted to smaller values, while correct predictions are still associated with high values, allowing a much better separability between these two types of prediction.

Based on this observation, we propose a novel approach for failure prediction with deep neural networks. We introduce a new confidence criteria based on the idea of using the TCP (section 2.1), for which we provide theoretical guarantees in the context of failure prediction. Since the true class is obviously unknown at test time, we introduce a method to learn a given target confidence criterion from data (section 2.2). We also discuss connections and differences between related works for failure prediction, in particular Bayesian deep learning and ensemble approaches, as well as recent approaches designing alternative criteria for failure prediction (section 2.3). We conduct extensive comparative experiments across various tasks, datasets and network architectures to validate the relevance of our proposed approach (section 3.2). Finally, a thorough analysis of our approach regarding the choice of loss function, criterion and learning scheme is presented in section 3.3.

## 2   Failure prediction by learning model confidence

We are interested in the problem of defining relevant confidence criteria for failure prediction with deep neural networks, in the context of classification. We also address semantic image segmentation, which can be seen as a pixel-wise classification problem, where a model outputs a dense segmentation mask with a predicted class assigned to each pixel. As such, all the following material is formulated for classification, and implementation details for segmentation are specified when necessary.

Let us consider a dataset $\mathcal{D}$ which consists of $N$ *i.i.d.* training samples $\mathcal{D} = \{(\mathbf{x}_i, y_i^*)\}_{i=1}^{N}$ where $\mathbf{x}_i \in \mathbb{R}^d$ is a $d$-dimensional feature and $y_i^* \in \mathcal{Y} = \{1, ..., K\}$ is its true class. We view a classification neural network as a probabilistic model: given an input $\mathbf{x}$, the network assigns a probabilistic predictive distribution $P(Y|\mathbf{w}, \mathbf{x})$ by computing the softmax output for each class $k$ and where $\mathbf{w}$

are the parameters of the network. From this predictive distribution, one can infer the class predicted by the model as $\hat{y} = \underset{k \in \mathcal{Y}}{\operatorname{argmax}} P(Y = k | \mathbf{w}, \mathbf{x})$.

During training, network parameters $\mathbf{w}$ are learned following a maximum likelihood estimation framework where one minimizes the Kullback-Leibler (KL) divergence between the predictive distribution and the true distribution. In classification, this is equivalent to minimizing the cross-entropy loss w.r.t. $\mathbf{w}$, which is the negative sum of the log-probabilities over positive labels:

$$\mathcal{L}_{\text{CE}}(\mathbf{w}; \mathcal{D}) = -\frac{1}{N} \sum_{i=1}^{N} y_i^* \log P(Y = y_i^* | \mathbf{w}, \mathbf{x}_i). \tag{1}$$

## 2.1 Confidence criterion for failure prediction

Instead of trying to improve the accuracy of a given trained model, we are interested in knowing if it can be endowed with the ability to recognize when its prediction may be wrong. A confidence criterion is a quantitative measure to estimate the confidence of the model prediction. The higher the value, the more certain the model about its prediction. As such, a suitable confidence criterion should correlate erroneous predictions with low values and successful predictions with high values. Here, we specifically focus on the ability of the confidence criterion to separate successful and erroneous predictions in order to distinguish them.

For a given input $\mathbf{x}$, a standard approach is to compute the softmax probability of the predicted class $\hat{y}$, that is the Maximum Class Probability: $\text{MCP}(\mathbf{x}) = \underset{k \in \mathcal{Y}}{\max} P(Y = k | \mathbf{w}, \mathbf{x}) = P(Y = \hat{y} | \mathbf{w}, \mathbf{x})$.

By taking the largest softmax probability, MCP leads to high confidence values both for errors and correct predictions, making it hard to distinguish them, as shown in Figure 1(a). On the other hand, when the model is misclassifying an example, the probability associated to the true class $y^*$ would be more likely close to a low value, reflecting the fact that the model made an error. Thus, we propose to consider the True Class Probability as a suitable confidence criterion for failure prediction:

$$\begin{aligned} \text{TCP}: \quad \mathbb{R}^d \times \mathcal{Y} &\to \mathbb{R} \\ (\mathbf{x}, \ y^*) &\to P(Y = y^* | \mathbf{w}, \mathbf{x}) \end{aligned} \tag{2}$$

**Theoretical guarantees.** With TCP, the following properties hold (see derivation in supplementary 1.1). Given an example $(\mathbf{x}, y^*)$,

- $\text{TCP}(\mathbf{x}, y^*) > 1/2 \Rightarrow \hat{y} = y^*$, *i.e.* the example is properly classified by the model,
- $\text{TCP}(\mathbf{x}, y^*) < 1/K \Rightarrow \hat{y} \neq y^*$, *i.e.* the example is wrongly classified by the model.

Within the range $[1/K, 1/2]$, there is no theoretical guarantee that correct and incorrect predictions will not overlap in terms of TCP. However, when using deep neural networks, we observe that the actual overlap area is extremely small in practice, as illustrated in Figure 1(b) on the CIFAR-10 dataset. One possible explanation comes from the fact that modern deep neural networks output overconfident predictions and therefore non-calibrated probabilities [13]. We provide consolidated results and analysis on this aspect in Section 3 and in the supplementary 1.2.

We also introduce a normalized variant of the TCP confidence criterion, which consists in computing the *ratio* between TCP and MCP:

$$\text{TCP}^r(\mathbf{x}, y^*) = \frac{P(Y = y^* | \mathbf{w}, \mathbf{x})}{P(Y = \hat{y} | \mathbf{w}, \mathbf{x})}. \tag{3}$$

The $\text{TCP}^r$ criterion presents stronger theoretical guarantees than TCP, since correct predictions will be, by design, assigned the value of 1, whereas errors will range in $[0, 1[$. On the other hand, learning this criterion may be more challenging since all correct predictions must match a single scalar value.

## 2.2 Learning TCP confidence with deep neural networks

Using TCP as confidence criterion on a model's output would be of great help when it comes to predicting failures. However, the true class $y^*$ of an output is obviously not available when

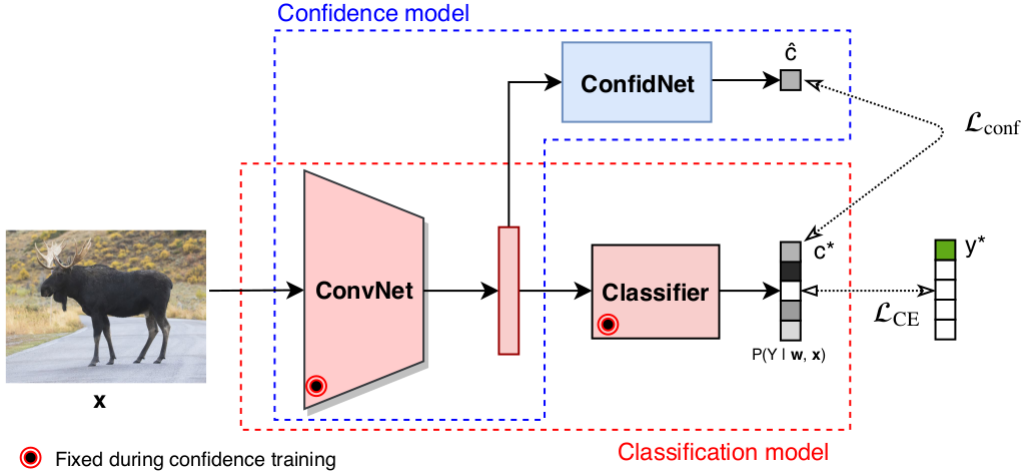

Figure 2: Our approach is based on two sub-networks. The classification model with parameters **w** is composed of a succession of convolutional and dense layers ('ConvNet') followed by a final dense layer with softmax activation. The confidence network, 'ConfidNet', builds upon features maps extracted by ConvNet, and is composed of a succession of layers which output a confidence score $\hat{c}(\mathbf{x}, \theta) \in [0, 1]$.

estimating confidence on test samples. Thus, we propose to learn TCP confidence $c^*(\mathbf{x}, y^*) = P(Y = y^* | \mathbf{w}, \mathbf{x})$ [1], our target confidence value. We introduce a confidence neural network, termed *ConfidNet*, with parameters $\theta$, which outputs a confidence prediction $\hat{c}(\mathbf{x}, \theta)$. During training, we seek $\theta$ such that $\hat{c}(\mathbf{x}, \theta)$ is close to $c^*(\mathbf{x}, y^*)$ on training samples (see Figure 2).

ConfidNet builds upon a classification neural network $M$, whose parameters **w** are preliminary learned using cross-entropy loss $\mathcal{L}_{\text{CE}}$ in (1). We are not concerned with improving model $M$'s accuracy. As a consequence, its classification layers (last fully connected layer and subsequent operations) will be fixed from now on.

**Confidence network design.** During initial classification training, model $M$ learns to extract increasingly complex features that are fed to the classification layers. To benefit from these rich representations, we build ConfidNet on top of them: ConfidNet passes these features through a succession of dense layers with a final sigmoid activation that outputs a scalar $\hat{c}(\mathbf{x}, \theta) \in [0, 1]$. Note that in semantic segmentation, models consist of fully convolutional networks where hidden representations are 2D feature maps. ConfidNet can benefit from this spatial information by replacing dense layers by $1 \times 1$ convolutions with adequate number of channels.

**Loss function.** Since we want to regress a score between 0 and 1, we use the $\ell_2$ loss to train ConfidNet:

$$\mathcal{L}_{\text{conf}}(\theta; \mathcal{D}) = \frac{1}{N} \sum_{i=1}^{N} (\hat{c}(\mathbf{x}_i, \theta) - c^*(\mathbf{x}_i, y_i^*))^2. \tag{4}$$

In the experimental part, we also tried more direct approaches for failure prediction such as a binary cross entropy loss (BCE) between the confidence network score and a incorrect/correct prediction target. We also tried implementing Focal loss [31], a BCE variant which focuses on hard examples. Finally, one can also see failure detection as a ranking problem where good predictions must be ranked before erroneous ones according to a confidence criterion. To this end, we also implemented a ranking loss [36, 7] applied locally on training batch inputs.

**Learning scheme.** Our complete confidence model, from input image to confidence score, shares its first encoding part ('ConvNet' in Fig.2) with the classification model $M$. The training of ConfidNet

starts by fixing entirely $M$ (freezing $\mathbf{w}$) and learning $\theta$ using loss (4). In a next step, we can then fine-tune the ConvNet encoder. However, as model $M$ has to remain fixed to compute similar classification predictions, we have now to decouple the feature encoders used for classification and confidence prediction respectively. We also deactivate dropout layers in this last training phase and reduce learning rate to mitigate stochastic effects that may lead the new encoder to deviate too much from the original one used for classification. Data augmentation can thus still be used.

## 2.3 Related works

Confidence estimation has already raised interest in the machine learning community over the past decade. Blatz *et al.* [3] introduce a method similar to our BCE baseline for confidence estimation in machine translation but their approach is not dedicated to training deep neural networks. Similarly, [42, 29] mention the use of bi-directional lattice RNN specifically designed for confidence estimation in speech recognition, whereas ConfidNet offers a model- and task-agnostic approach which can be plugged into any deep neural network. Post-hoc selective classification methods [11] identify a threshold over a confidence-rate function (e.g., *MCP*) to satisfy a user-specified risk level, whereas we focus here on relative metrics. Recently, Hendricks *et al.* [17] established a standard baseline for deep neural networks which relies on MCP retrieved from softmax distribution. As stated before, MCP presents several limits regarding both failure prediction and out-of-distribution detection as it outputs high confidence values. This limit is alleviated in our TCP criterion which also provides some interesting theoretical guarantees regarding confidence threshold.

In [20], Jiang *et al.* propose a new confidence measure, 'Trust Score', which measures the agreement between the classifier and a modified nearest-neighbor classifier on the test examples. More precisely, the confidence criterion used in Trust Score [20] is the ratio between the distance from the sample to the nearest class different from the predicted class and the distance to the predicted class. One clear drawback of this approach is its lack of scalability, since computing nearest neighbors in large datasets is extremely costly in both computation and memory. Another more fundamental limitation related to the Trust Score itself is that local distance computation becomes less meaningful in high dimensional spaces [2], which is likely to negatively affect performances of this method. In contrast, ConfidNet is based on a training approach which learns a sub-manifold in the error/success space, which is arguably less prone to the curse of dimensionality and, therefore, facilitate discrimination between these classes.

Bayesian approaches for uncertainty estimation in neural networks gained a lot of attention recently, especially due to the elegant connection between efficient stochastic regularization techniques, *e.g.* dropout [10], and variational inference in Bayesian neural networks [10, 9, 4, 21, 22]. Gal and Ghahramani proposed in [10] using Monte Carlo Dropout (MCDropout) to estimate the *posterior* predictive network distribution by sampling several stochastic network predictions. When applied to regression, the predictive distribution uncertainty can be summarized by computing statistics, *e.g.* variance. When using MCDropout for uncertainty estimation in classification tasks, however, the predictive distribution is averaged to a point-wise softmax estimate before computing standard uncertainty criteria, *e.g.* entropy or variants such as mutual information. It is worth mentioning that these entropy-based criteria measure the softmax output dispersion, where the uniform distribution has maximum entropy. It is not clear how well these dispersion measures are adapted for distinguishing failures from correct predictions, especially with deep neural networks which output overconfident predictions [13]: for example, it might be very challenging to discriminate a peaky prediction corresponding to a correct prediction from an incorrect overconfident one. We illustrate this issue in section 3.2.

In tasks closely related to failure prediction, other approaches also identified the issue of MCP regarding high confidence predictions [17, 30, 26, 28, 13, 40]. Guo *et al.* [13], for confidence calibration, and Liang *et al.* [30], for out-of-distribution detection, proposed to use temperature scaling to mitigate confidence values. However, this doesn't affect the ranking of the confidence score and therefore the separability between errors and correct predictions. DeVries *et al.* [6] share with us the same purpose of learning confidence in neural networks. Their work differs by focusing on out-of-distribution detection and learning jointly a distribution confidence score and classification probabilities. In addition, they use predicted confidence score to interpolate output probabilities and target whereas we specifically define TCP, a criterion suited for failure prediction.

Lakshminarayanan *et al.* [26] propose an alternative to Bayesian neural networks by leveraging ensemble of neural networks to produce well-calibrated uncertainty estimates. Part of their approach relies on using a proper scoring rule as training criterion. It is interesting to note that our TCP criterion corresponds actually to the exponential cross-entropy loss value of a model prediction, which is a proper scoring rule in the case of multi-class classification.

## 3 Experiments

In this section, we evaluate our approach to predict failure in both classification and segmentation settings. First, we run comparative experiments against state-of-the-art confidence estimation and Bayesian uncertainty estimation methods on various datasets. These results are then completed by a thorough analysis of the influence of the confidence criterion, the training loss and the learning scheme in our approach. Finally, we provide a few visualizations to get additional insight into the behavior of our approach. Our code is available at https://github.com/valeoai/ConfidNet.

### 3.1 Experimental setup

**Datasets.** We run experiments on image datasets of varying scale and complexity: MNIST [27] and SVHN [39] datasets provide relatively simple and small ($28 \times 28$) images of digits (10 classes). CIFAR-10 and CIFAR-100 [24] propose more complex object recognition tasks on low resolution images. We also report experiments for semantic segmentation on CamVid [5], a standard road scene dataset. Further details about these datasets, as well as on architectures, training and metrics can be found in supplementary 2.1.

**Network architectures.** The classification deep architectures follow those proposed in [20] for fair comparison. They range from small convolutional networks for MNIST and SVHN to larger VGG-16 architecture for the CIFAR datasets. We also added a multi-layer perceptron (MLP) with 1 hidden layer for MNIST to investigate performances on small models. For CamVid, we implemented a SegNet semantic segmentation model, following [21].

Our confidence prediction network, ConfidNet, is attached to the penultimate layer of the classification network. It is composed of a succession of 5 dense layers. Variants of this architecture have been tested, leading to similar performances (see supplementary 2.2 for more details). Following our specific learning scheme, we first train ConfidNet layers before fine-tuning the duplicate ConvNet encoder dedicated to confidence estimation. In the context of semantic segmentation, we adapt ConfidNet by making it fully convolutional.

**Evaluation metrics.** We measure the quality of failure prediction following the standard metrics used in the literature [17]: **AUPR-Error**, **AUPR-Success**, **FPR at 95% TPR** and **AUROC**. We will mainly focus on AUPR-Error, which computes the area under the Precision-Recall curve using errors as the positive class.

### 3.2 Comparative results on failure prediction

To demonstrate the effectiveness of our method, we implemented competitive confidence and uncertainty estimation approaches including Maximum Class Probability (MCP) as a baseline [17], Trust Score [20], and Monte-Carlo Dropout (MCDropout) [10]. For Trust Score, we used the code provided by the authors[2]. Further implementation details and parameter settings are available in the supplementary 2.1.

Comparative results are summarized in Table 1. First of all, we observe that our approach outperforms baseline methods in every setting, with a significant gap on small models/datasets. This confirms both that TCP is an adequate confidence criterion for failure prediction and that our approach ConfidNet is able to learn it. TrustScore method also presents good results on small datasets/models such as MNIST where it improved baseline. While ConfidNet still performs well on more complex datasets, Trust Score's performance drops, which might be explained by high dimensionality issues with distances as mentioned in section 2.3. For its application to semantic segmentation where each training pixel is a 'neighbor', computational complexity forced us to reduce drastically the number of training neighbors and of test samples. We sampled randomly in each train and test image a

Table 1: Comparison of failure prediction methods on various datasets. All methods share the same classification network. Note that for MCDropout, test accuracy is averaged over random sampling. All values are percentages.

| Dataset | Model | FPR-95%-TPR | AUPR-Error | AUPR-Success | AUC |
|---|---|---|---|---|---|
| **MNIST** MLP | Baseline (MCP) [17] | 14.87 | 37.70 | 99.94 | 97.13 |
| | MCDropout [10] | 15.15 | 38.22 | 99.94 | 97.15 |
| | TrustScore [20] | 12.31 | 52.18 | 99.95 | 97.52 |
| | ConfidNet (Ours) | **11.79** | **57.37** | **99.95** | **97.83** |
| **MNIST** Small ConvNet | Baseline (MCP) [17] | 5.56 | 35.05 | 99.99 | 98.63 |
| | MCDropout [10] | 5.26 | 38.50 | 99.99 | 98.65 |
| | TrustScore [20] | 10.00 | 35.88 | 99.98 | 98.20 |
| | ConfidNet (Ours) | **3.33** | **45.89** | **99.99** | **98.82** |
| **SVHN** Small ConvNet | Baseline (MCP) [17] | 31.28 | 48.18 | 99.54 | 93.20 |
| | MCDropout [10] | 36.60 | 43.87 | 99.52 | 92.85 |
| | TrustScore [20] | 34.74 | 43.32 | 99.48 | 92.16 |
| | ConfidNet (Ours) | **28.58** | **50.72** | **99.55** | **93.44** |
| **CIFAR-10** VGG16 | Baseline (MCP) [17] | 47.50 | 45.36 | 99.19 | 91.53 |
| | MCDropout [10] | 49.02 | 46.40 | **99.27** | 92.08 |
| | TrustScore [20] | 55.70 | 38.10 | 98.76 | 88.47 |
| | ConfidNet (Ours) | **44.94** | **49.94** | 99.24 | **92.12** |
| **CIFAR-100** VGG16 | Baseline (MCP) [17] | 67.86 | 71.99 | 92.49 | 85.67 |
| | MCDropout [10] | 64.68 | 72.59 | **92.96** | 86.09 |
| | TrustScore [20] | 71.74 | 66.82 | 91.58 | 84.17 |
| | ConfidNet (Ours) | **62.96** | **73.68** | 92.68 | **86.28** |
| **CamVid** SegNet | Baseline (MCP) [17] | 63.87 | 48.53 | 96.37 | 84.42 |
| | MCDropout [10] | 62.95 | 49.35 | 96.40 | 84.58 |
| | TrustScore [20] | | 20.42 | 92.72 | 68.33 |
| | ConfidNet (Ours) | **61.52** | **50.51** | **96.58** | **85.02** |

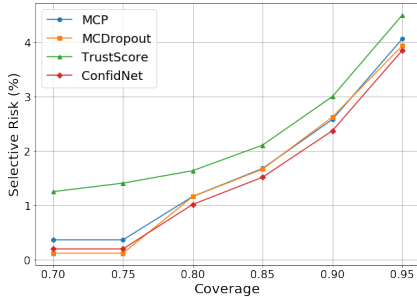

(a) CIFAR-10

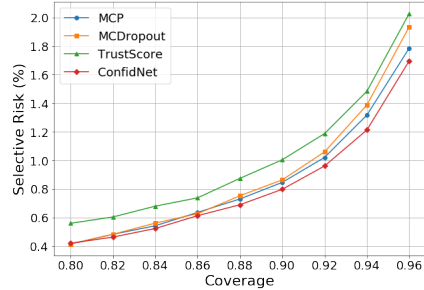

(b) SVHN

Figure 3: Risk-coverage curves. 'Selective risk' ($y$-axis) represents the percentage of errors in the remaining test set for a given coverage percentage.

small percentage of pixels to compute TrustScore. ConfidNet, in contrast, is as fast as the original segmentation network.

We also improve state-of-art performances from MCDropout. While MCDropout leverages ensembling based on dropout layers, taking as confidence measure the entropy on the average softmax distribution may not be always adequate. In Figure 4, we show side-by-side two samples with a similar distribution entropy. Left image is misclassified while right one enjoys a correct prediction. Entropy is a symmetric measure in regards to class probabilities: a correct prediction with $[0.65, 0.35]$ distribution is evaluated as confident as an incorrect one with $[0.35, 0.65]$ distribution. In contrast, our approach can discriminate an incorrect from a correct prediction despite both having similarly spread distributions.

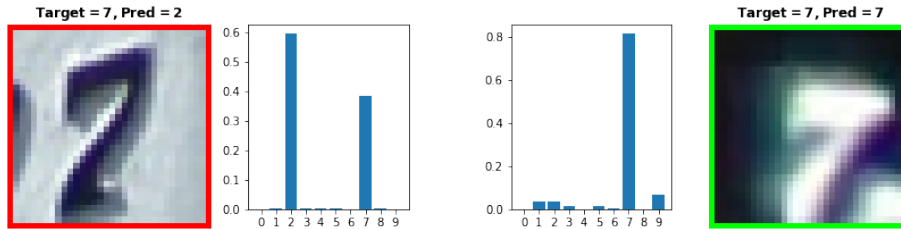

(a) MCP=0.596, MCDropout=-0.787, *ConfidNet*=0.449      (b) MCP=0.816, MCDropout=-0.786, *ConfidNet*=0.894

Figure 4: Illustrating the limits of MCDropout with entropy as confidence estimation on SVHN test samples. Red-border image (a) is misclassified by the classification model; green-border image (b) is correctly classified. Prediction exhibit similar high entropy in both cases. For each sample, we provide a plot of their softmax predictive distribution.

Risk-coverage curves [8, 11] depicting the performance of ConfidNet and other baselines for CIFAR-10 and SVHN datasets appear in Figure 3. 'Coverage' corresponds to the probability mass of the non-rejected region after using a threshold as selection function [11]. For both datasets, ConfidNet presents a better coverage potential for each selective risk that a user can choose beforehand. In addition, we can see that the improvement is more pronounced at high coverage rates - *e.g.* in $[0.8; 0.95]$ for CIFAR-10 (Fig. 3a) and in $[0.86; 0.96]$ for SVHN (Fig. 3b) - which highlights the capacity of ConfidNet to identify successfully critical failures.

## 3.3 Effect of learning variants

We first evaluate the effect of fine-tuning ConvNet in our approach. Without fine-tuning, ConfidNet already achieves significant improvements w.r.t. baseline, as shown in Table 2. By allowing subsequent fine-tuning as described in

Table 2: Effect of learning scheme on AUPR-Error

|  | MNIST SmallConvNet | CIFAR-100 VGG-16 |
|---|---|---|
| Confidence training | 43.94% | 72.68% |
| + Fine-tuning ConvNet | 45.89% | 73.68% |

section 2.2, ConfidNet performance is further boosted in every setting, around 1-2%. Note that using a vanilla fine-tuning without deactivating dropout layers did not bring any improvement.

Given the small number of errors available due to deep neural network over-fitting, we also experimented with training ConfidNet on a hold-out dataset. We report results on all datasets in Table 3 for validation sets with 10% of samples. We observe a general performance drop when using a validation set for training TCP confidence. The drop is especially pronounced for small datasets (MNIST), where models reach >97% train and val accuracies. Consequently, with a high accuracy and a small validation set, we do not get a larger absolute number of errors using val set compared to train set. One solution would be to increase validation set size but this would damage model's prediction performance. By contrast, we take care with our approach to base our confidence estimation on models with levels of test predictive performance that are similar to those of baselines. On CIFAR-100, the gap between train accuracy and val accuracy is substantial (95.56% vs. 65.96%), which may explain the slight improvement for confidence estimation using val set (+0.17%). We think that training ConfidNet on val set with models reporting low/middle test accuracies could improve the approach.

Table 3: Comparison between training ConfidNet on train set or on validation set

| AUPR-Error (%) | MNIST MLP | MNIST SmallConvNet | SVHN SmallConvNet | CIFAR-10 VGG-16 | CIFAR-100 VGG-16 | CamVid SegNet |
|---|---|---|---|---|---|---|
| ConfidNet (using train set) | 57.34% | 43.94% | 50.72% | 49.94% | 73.68% | 50.28% |
| ConfidNet (using val set) | 33.41% | 34.22% | 47.96% | 48.93% | 73.85% | 50.15% |

On Table 4, we compare training ConfidNet with MSE loss to binary classification cross-entropy loss (BCE). Even though BCE specifically addresses the failure prediction task, we

observe that it achieves lower performances on CIFAR-10 and CamVid datasets. Focal loss and ranking loss were also tested and presented similar results (see supplementary 2.3).

We intuitively think that TCP regularizes training by providing more fine-grained information about the quality of the classifier regarding a sample's prediction. This is especially important in the difficult

Table 4: Effect of loss and normalized criterion on AUPR-Error

| Dataset | TCP | Loss BCE | Criterion $TCP^r$ |
|---|---|---|---|
| **CIFAR-10** | 49.94% | 47.95% | 48.78% |
| **CamVid** | 50.51% | 48.96% | 51.35% |

learning configuration where only very few error samples are available due to the good performance of the classifier. We also evaluate the impact of regression to the normalized criterion $TCP^r$: performance is lower than the one of TCP on small datasets such as CIFAR-10 where few errors are present, but higher on larger datasets such as CamVid where each pixel is a sample. This emphasizes once again the complexity of incorrect/correct classification training.

### 3.4 Qualitative assessments

In this last subsection, we provide an illustration on CamVid (Figure 5) to better understand our approach for failure prediction. Compared to MCP baseline, our approach produces higher confidence scores for correct pixel predictions and lower ones on erroneously predicted pixels, which allow an user to better detect errors area in semantic segmentation.

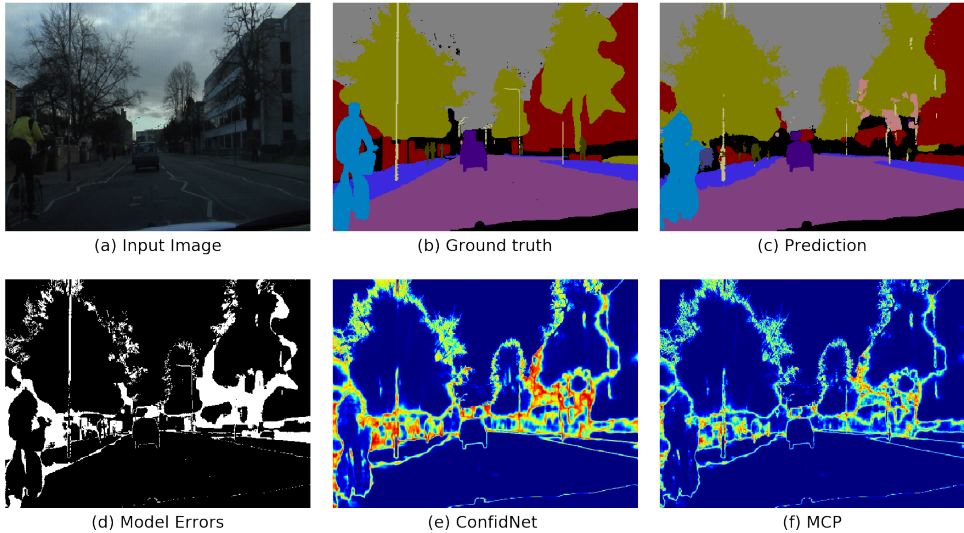

Figure 5: Comparison of inverse confidence (uncertainty) map between ConfidNet (e) and MCP (f) on one CamVid scene. The top row shows the input image (a) with its ground-truth (b) and the semantic segmentation mask (c) predicted by the original classification model. The error map associated to the predicted segmentation is shown in (d), with erroneous predictions flagged in white. ConfidNet (55.53% AP-Error) allows a better prediction of these errors than MCP (54.69% AP-Error).

## 4   Conclusion

In this paper, we defined a new confidence criterion, TCP, which provides both theoretical guarantees and empirical evidences to address failure prediction. We proposed a specific method to learn this criterion with a confidence neural network built upon a classification model. Results showed a significant improvement from strong baselines on various classification and semantic segmentation datasets, which validate the effectiveness of our approach. Further works involve exploring methods to artificially generate errors, such as in adversarial training. ConfidNet could also be applied for uncertainty estimation in domain adaptation [45, 14] or in multi-task learning [23, 38].

## Footnotes

[1]or its normalized variant $\text{TCP}^r(\mathbf{x}, y^*)$.

[2]https://github.com/google/TrustScore

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
