[Supplementary Material]

# Addressing Failure Prediction
# by Learning Model Confidence
## *Supplementary Material*

**Charles Corbière**
charles.corbiere@valeo.com

**Nicolas Thome**
nicolas.thome@cnam.fr

**Avner Bar-Hen**
avner@cnam.fr

**Matthieu Cord**
matthieu.cord@lip6.fr

**Patrick Pérez**
patrick.perez@valeo.com

# 1 True Class Probability (TCP) criterion

## 1.1 Proof of TCP theoretical guarantees

Let $K$ be the number of labels and $\mathbf{x} \in \mathbb{R}^D$ a sample with its associated label $y^* \in \mathcal{Y}$ such that $\mathrm{TCP}(\mathbf{x}, y^*) > \frac{1}{2}$. Starting from the definition of *TCP* we have:

$$\mathrm{TCP}(\mathbf{x}, y^*) = P(Y = y^*|\mathbf{w}, \mathbf{x}) > \frac{1}{2} \tag{1}$$

$$\iff 1 - \sum_{k \in \mathcal{Y}, k \neq y^*} P(Y = k|\mathbf{w}, \mathbf{x}) > \frac{1}{2} \tag{2}$$

$$\iff \sum_{k \in \mathcal{Y}, k \neq y^*} P(Y = k|\mathbf{w}, \mathbf{x}) < \frac{1}{2}. \tag{3}$$

Since probabilities are positive, we obtain that $\forall k \neq y^*$, $P(Y = k|\mathbf{w}, \mathbf{x}) < \frac{1}{2} < P(Y = y^*|\mathbf{w}, \mathbf{x})$. Denoting $\hat{y}$ the class predicted by the network, we have $\hat{y} = \arg\max_k P(Y = k|\mathbf{w}, \mathbf{x})$. Hence $\hat{y} = y^*$.

In the same way, for $\mathbf{x} \in \mathbb{R}^D$ and $y^* \in \mathcal{Y}$, such that $\mathrm{TCP}(\mathbf{x}, y^*) < \frac{1}{K}$, we have:

$$P(Y = y^*|\mathbf{w}, \mathbf{x}) < \frac{1}{K} \tag{4}$$

$$\iff 1 - \sum_{k \in \mathcal{Y}, k \neq y^*} P(Y = k|\mathbf{w}, \mathbf{x}) < \frac{1}{K} \tag{5}$$

$$\iff \sum_{k \in \mathcal{Y}, k \neq y^*} P(Y = k|\mathbf{w}, \mathbf{x}) > \frac{K-1}{K}. \tag{6}$$

If the model correctly classifies this sample, i.e. $\hat{y} = y^*$, then $\forall k \neq y^*$, $P(Y = y^*|\mathbf{w}, \mathbf{x}) \geq P(Y = k|\mathbf{w}, \mathbf{x})$. We have then:

$$\sum_{k \in \mathcal{Y}, k \neq y^*} P(Y = k|\mathbf{w}, \mathbf{x}) \leq (K-1)P(Y = y^*|\mathbf{w}, \mathbf{x}) \leq \frac{K-1}{K}, \tag{7}$$

which contradicts Equation (6). Hence, there exists at least one $k$ such that $P(Y = k|\mathbf{w}, \mathbf{x}) > P(Y = y^*|\mathbf{w}, \mathbf{x})$, which results in $\hat{y} \neq y^*$.

## 1.2 Empirical error and success distributions

Figure 1: Distribution plot for MNIST MLP

(a) Maximum Class Probability  (b) Our Proposal (True Class Probability)

| Model | # Errors | | | # Successes | | | AUPR-Error | AUPR-Success | AUC |
|---|---|---|---|---|---|---|---|---|---|
| | $> 1/K$ | $[\frac{1}{K}, \frac{1}{2}]$ | $>1/2$ | $< 1/K$ | $[\frac{1}{K}, \frac{1}{2}]$ | $> 1/2$ | | | |
| MCP | 0 | 25 | 170 | 0 | 28 | 9777 | 37.70% | 99.94% | 97.13% |
| TCP | 81 | 114 | 0 | 0 | 28 | 9777 | 98.77% | 100.00% | 99.98% |

Figure 2: Distribution plot for MNIST Small ConvNet

(a) Maximum Class Probability  (b) Our Proposal (True Class Probability)

| Model | # Errors | | | # Successes | | | AUPR-Error | AUPR-Success | AUC |
|---|---|---|---|---|---|---|---|---|---|
| | $> 1/K$ | $[\frac{1}{K}, \frac{1}{2}]$ | $>1/2$ | $< 1/K$ | $[\frac{1}{K}, \frac{1}{2}]$ | $> 1/2$ | | | |
| *MCP* | 0 | 8 | 82 | 0 | 11 | 9899 | 35.05% | 99.99% | 98.63% |
| *TCP* | 32 | 58 | 0 | 0 | 11 | 9899 | 99.41% | 100.00% | 99.41% |

Figure 3: Distribution plot for SVHN Small ConvNet

(a) Maximum Class Probability  (b) Our Proposal (True Class Probability)

| Model | # Errors | | | # Successes | | | AUPR-Error | AUPR-Success | AUC |
|---|---|---|---|---|---|---|---|---|---|
| | $> 1/K$ | $[\frac{1}{K}, \frac{1}{2}]$ | $>1/2$ | $< 1/K$ | $[\frac{1}{K}, \frac{1}{2}]$ | $> 1/2$ | | | |
| *MCP* | 0 | 329 | 857 | 0 | 206 | 24640 | 48.18% | 99.54% | 93.20% |
| *TCP* | 500 | 686 | 0 | 0 | 206 | 24640 | 98.93% | 100.00% | 99.95% |

Figure 4: Distribution plot for CIFAR-10 VGG16

(a) Maximum Class Probability  (b) Our Proposal (True Class Probability)

| Model | # Errors | | | # Successes | | | AUPR-Error | AUPR-Success | AUC |
|---|---|---|---|---|---|---|---|---|---|
| | $> 1/K$ | $[\frac{1}{K}, \frac{1}{2}]$ | $>1/2$ | $< 1/K$ | $[\frac{1}{K}, \frac{1}{2}]$ | $> 1/2$ | | | |
| MCP | 0 | 52 | 729 | 0 | 33 | 9186 | 45.36% | 99.19% | 91.56% |
| TCP | 469 | 312 | 0 | 0 | 33 | 9186 | 99.77% | 100.00% | 99.98% |

Figure 5: Distribution plot for CIFAR-100 VGG16

(a) Maximum Class Probability  (b) Our Proposal (True Class Probability)

| Model | # Errors | | | # Successes | | | AUPR-Error | AUPR-Success | AUC |
|---|---|---|---|---|---|---|---|---|---|
| | $> 1/K$ | $[\frac{1}{K}, \frac{1}{2}]$ | $>1/2$ | $< 1/K$ | $[\frac{1}{K}, \frac{1}{2}]$ | $> 1/2$ | | | |
| MCP | 0 | 603 | 2801 | 0 | 118 | 6478 | 71.99% | 92.49% | 85.67% |
| TCP | 2724 | 680 | 0 | 0 | 118 | 6478 | 99.91% | 99.98% | 99.91% |

Figure 6: Distribution plot for CamVid SegNet

(a) Maximum Class Probability  (b) Our Proposal (True Class Probability)

| Model | # Errors | | | # Successes | | | AUPR-Error | AUPR-Success | AUC |
|---|---|---|---|---|---|---|---|---|---|
| | $> 1/K$ | $[\frac{1}{K}, \frac{1}{2}]$ | $>1/2$ | $< 1/K$ | $[\frac{1}{K}, \frac{1}{2}]$ | $> 1/2$ | | | |
| MCP | 0 | 401,573 | 55,506,172 | 0 | 188,128 | 34,166,526 | 48.53% | 96.37% | 84.42% |
| TCP | 41,84,875 | 1,722,871 | 0 | 0 | 188,128 | 34,166,526 | 99.92% | 100.00% | 99.99% |

## 2   Experiments

### 2.1   Implementation details

**Datasets.**   We run experiments on image datasets of varying scale and complexity: MNIST [7] and SVHN [8] datasets provide relatively simple and small ($28 \times 28$) images of digits (10 classes). They are splited in 60,000 training samples and 10,000 testing samples. CIFAR-10 and CIFAR-100 [6] bring more complexity to classify low resolution images. In each dataset, we further keep 10% of training samples as a validation dataset. We also report experiments for semantic segmentation on CamVid [1], a standard road scene dataset. Images are resized to $360 \times 480$ pixels and are segmented according to 11 classes such as 'road', 'building', 'car' or 'pedestrian'.

**Classification network.**   For each dataset, we use standard neural network architectures as classifiers. We re-implemented in PyTorch [9] network architectures proposed in [4] for fair comparison. They range from small convolutional networks for MNIST[1] and SVHN[2] to VGG-16 architectures[3] for CIFAR datasets. We also added a multi-layer perceptron (MLP) with 1 hidden layer of size 100 for MNIST dataset in order to investigate performances on small models. Finally, we implemented a SegNet following [5]. All models are trained in a standard way with a cross-entropy loss and a SGD optimizer with a learning rate of $10^{-3}$, a momentum of 0.9 and a weight decay of $10^{-4}$. Number of training epochs depends on the dataset considered, varying from 100 epochs on MNIST to 250 epochs on CIFAR-100. As we're looking to compute Monte Carlo samples following [2], we also include dropout layers. Best model is selected on validation set accuracy.

**ConfidNet training.**   We train ConfidNet for 500 epochs with Adam optimizer with learning rate $10^{-4}$, dropout and same data augmentation used in classification training. We select best model based on AUPR-Error on validation dataset. To specifically fine-tune the encoder used for ConfidNet, we decoupled the encoder from original ConvNet and allow back-propagation through it. Training is completed on very few epochs based on previous best model, using Adam optimizer with learning rate $10^{-7}$ or $10^{-8}$ and no dropout to mitigate stochastic effects that may lead the new encoder to deviate too much from the original one used for classification. Once again, best model is selected on val set AUPR-Error metrics.

**Evaluation metrics.**

1. **FPR at 95% TPR** measures the False Positive Rate (FPR) when the True Positive Rate (TPR) is equal to 95%. True Positive Rate can be computed by $\mathrm{TPR} = \mathrm{TP}/(\mathrm{TP} + \mathrm{FN})$, where TP and FN denote numbers of true positives and false negatives respectively. The False Positive Rate can be computed by $\mathrm{FPR} = \mathrm{FP}/(\mathrm{FP} + \mathrm{TN})$, where FP and TN denote the number of false positives and true negatives respectively. This metric can be interpreted as the probability that an error is misclassified as a correct prediction when the True Positive Rate (TPR) is as high as 95%.

2. **AUROC** measures the Area Under the Receiver Operating Characteristic curve (AUROC). The ROC curve is a graph showing True Positive Rate versus False Positive Rate. This metric is a threshold-independent performance evaluation, such as AUPR. It can be interpreted as the probability that a positive example has a greater prediction score than a negative example.

3. **AUPR** measures the Area Under the Precision-Recall (PR) curve. The PR curve is a graph showing precision $= \mathrm{TP}/(\mathrm{TP} + \mathrm{FP})$ versus recall $= \mathrm{TP}/(\mathrm{TP} + \mathrm{FN})$. In our tests, AUPR-Success indicates that correct predictions are used as the positive class, while AUPR-Error indicates that errors are used as the positive class. As we specifically want to detect failures, AUPR-Error constitutes the primary metrics to assess performances.

**Other baseline details**   For TrustScore [4], we add parallel processing when computing distances for each class to speed up inference. This parallelization does not alter the algorithm nor its performances. Specifically for semantic segmentation with CamVid, each image contains 172,800

pixels. Even though CamVid remains a small dataset (367 training images, 101 validation images, 233 test images) compared to other semantic segmentation datasets, computation complexity forced us to drastically reduce the number of training neighbors and the number of test samples. We randomly sample in each train and test image a small percentage of pixels to compute a proxy.

For MC-Dropout [2], we use the same model than baseline (which already includes dropout layers) and we sample 100 times from the classification model at test time keeping dropout layers activated. We then compute the average softmax probability over all samples to conduct Monte Carlo integration. Model uncertainty is estimated following [2] by calculating the entropy of the averaged probability vector across the class dimension.

## 2.2 Classification accuracies

Most neural networks used in our experiments tend to overfit. On small datasets such as MNIST and SVHN, convolutional neural networks already achieve nearly perfect accuracy on test set, above 96%, which leaves very few errors available. We provide on Table 1 accuracies on training, validation and test set.

Table 1: Train, val and test accuracies for each model.

|  | MNIST MLP | MNIST SmallConvNet | SVHN Small ConvNet | CIFAR-10 VGG-16 | CIFAR-100 VGG-16 | CamVid SegNet |
|---|---|---|---|---|---|---|
| Train accuracy | 98.32% | 98.94% | 95.06% | 98.69% | 95.55% | 96.69% |
| Val accuracy | 97.95% | 99.03% | 96.56% | 99.80% | 66.96% | 91.72% |
| Test accuracy | 98.05% | 99.10% | 95.44% | 92.19% | 65.96% | 85.33% |

## 2.3 Effect of ConfidNet architecture

We experiment different ConfidNet architectures on the SVHN dataset, varying the number of layers. Except for first and last layers, whose dimensions respectively depend on input and output size, each layer presents the same number of units (400). On Fig 7, we observe that starting from 3 layers, ConfidNet already improves baseline performance.

Figure 7: Influence of the number of layers used in ConfidNet on SVHN test set.

## 2.4 Effect of learning variants

Table 2: Effect of learning scheme on AUPR-Error

|  | MNIST MLP | MNIST SmallConvNet | SVHN Small ConvNet | CIFAR-10 VGG-16 | CIFAR-100 VGG-16 | CamVid SegNet |
|---|---|---|---|---|---|---|
| Confidence training | 57.34% | 43.94% | 50.43% | 46.44% | 72.68% | 50.12% |
| + Fine-tuning ConvNet | 57.37% | 45.89% | 50.72% | 49.94% | 73.68% | 50.51% |

Table 3: Effect of loss and criterion on SVHN and CamVid

| Dataset | Loss | FPR (95% TPR) | AUPR-Error | AUPR-Success | AUC |
|---|---|---|---|---|---|
| **SVHN**<br>Small ConvNet | TCP | **28.58%** | **50.72%** | **99.55%** | **93.44%** |
| | BCE | 29.34% | 50.00% | 99.52% | 92.76% |
| | Focal | 28.67% | 49.96% | 99.53% | 93.01% |
| | Ranking | 31.04% | 48.11% | 99.55% | 92.90% |
| | $TCP^r$ | 30.19% | 47.04% | 99.53% | 93.12% |
| **CIFAR-10**<br>VGG-16 | TCP | **44.94%** | **49.94%** | 99.24% | 92.12% |
| | BCE | 45.20% | 47.95% | 99.19% | 91.94% |
| | Focal | 45.20% | 47.76% | 99.22% | 91.93% |
| | Ranking | 46.99% | 44.04% | 99.19% | 91.49% |
| | $TCP^r$ | 44.43% | 48.78% | **99.25%** | **92.19%** |
| **CamVid**<br>SegNet | TCP | 61.52% | 50.51% | 96.58% | 85.02% |
| | BCE | 61.68% | 48.96% | 96.05% | 83.41% |
| | Focal | 61.64% | 49.05% | 96.49% | 84.09% |
| | $TCP^r$ | **60.41%** | **51.35%** | **96.58%** | **85.18%** |

## 2.5 Effect on calibration

We empirically observed that ConfidNet tend to lower over-confident predictions which happen to be errors. As a side experiment, we thus study whether using ConfidNet as confidence estimation improve calibration of deep neural networks. We report the Expected Calibration Error (ECE) which is an approximate measure of miscalibration between confidence and accuracy [3]. Table 4 sums up our results.

Table 4: Calibration results for ConfidNet

| ECE (%) | MNIST<br>MLP | MNIST<br>SmallConvNet | SVHN<br>SmallConvNet | CIFAR-10<br>VGG-16 | CIFAR-100<br>VGG-16 | CamVid<br>SegNet |
|---|---|---|---|---|---|---|
| Baseline | 0.37% | **0.20%** | **0.50%** | 4.48% | 22.37% | 9.65% |
| ConfidNet | 0.66% | 0.30% | 1.11% | 3.45% | 15.61% | 7.57% |
| Baseline + T. Scaling | **0.20%** | 0.69% | 1.30% | **2.88%** | **5.16%** | **4.77%** |

We observe that ConfidNet presents equivalent or better ECE results than baseline, mostly pronounced on complex datasets such as CIFAR-10, CIFAR-100 and CamVid. On MNIST and SVHN, baseline already presented small ECE results. These results confirm our intuition about the capacity of ConfidNet to address over confident predictions, even though it has not been designed for. Nevertheless, dedicated methods such as temperature scaling used in [3] remain preferred for calibrate deep neural networks.

## 2.6 Risk-coverage curves

Added to plot provided in the paper for VGG-16 on CIFAR-10 dataset, we include here plots for the remaining datasets: MLP on MNIST (Fig 8a), Small ConvNet on MNIST (Fig 8b), Small ConvNet on SVHN (Fig 8c) and VGG-16 on CIFAR-100 (Fig 8e).

For further details, we also provide quantitative table for risk-coverage curves for SVHN (Fig 5) and CIFAR-10 (Fig 6) datasets.

(a) MLP

(b) MNIST

(c) SVHN

(d) CIFAR-10

(e) CIFAR-100

Figure 8: Risk-coverage curves. Selective risk represents the percentage of errors in the remaining test set for a given coverage percentage.

Table 5: Selective risk for various coverage rates on SVHN

| Coverage | ConfidNet risk | MCP risk | % improvement | MCDropout risk | % improvement | TrustScore risk | % improvement |
|---|---|---|---|---|---|---|---|
| 1.00 | **4.55** | 4.55 | 0.00 | 4.55 | 0.00 | 4.55 | 0.00 |
| 0.98 | **2.46** | 2.63 | 6.37 | 2.80 | 12.09 | 2.87 | 14.12 |
| 0.96 | **1.70** | 1.79 | 4.98 | 1.94 | 12.31 | 2.03 | 16.33 |
| 0.94 | **1.21** | 1.32 | 7.92 | 1.39 | 12.61 | 1.48 | 18.25 |
| 0.92 | **0.96** | 1.02 | 5.88 | 1.06 | 9.30 | 1.19 | 19.09 |
| 0.90 | **0.80** | 0.85 | 5.71 | 0.86 | 7.64 | 0.99 | 19.49 |
| 0.88 | **0.69** | 0.73 | 5.53 | 0.75 | 8.36 | 0.87 | 21.09 |
| 0.86 | **0.61** | 0.64 | 3.63 | 0.63 | 2.21 | 0.74 | 16.90 |
| 0.84 | **0.52** | 0.54 | 3.49 | 0.56 | 6.74 | 0.68 | 22.96 |
| 0.82 | **0.46** | 0.48 | 4.03 | 0.48 | 4.00 | 0.60 | 23.27 |
| 0.80 | 0.42 | 0.42 | -1.16 | **0.42** | -1.19 | 0.56 | 24.92 |

Table 6: Selective risk for various coverage rates on CIFAR-10

| Coverage | ConfidNet risk | MCP risk | % improvement | MCDropout risk | % improvement | TrustScore risk | % improvement |
|---|---|---|---|---|---|---|---|
| 1.00 | **7.80** | 7.80 | 0.00 | 7.80 | 0.00 | 7.80 | 0.00 |
| 0.95 | **3.85** | 4.06 | 5.18 | 3.94 | 2.14 | 4.54 | 15.22 |
| 0.90 | **2.38** | 2.58 | 8.00 | 2.63 | 9.59 | 2.96 | 19.68 |
| 0.85 | **1.52** | 1.68 | 9.58 | 1.67 | 8.90 | 2.06 | 26.26 |
| 0.80 | **1.02** | 1.17 | 12.49 | 1.17 | 12.44 | 1.63 | 37.27 |
| 0.75 | 0.20 | 0.37 | 45.40 | **0.12** | -62.54 | 1.38 | 85.45 |
| 0.70 | 0.20 | 0.37 | 45.40 | **0.12** | -62.54 | 1.33 | 84.87 |

## Footnotes

[1]https://github.com/EN10/KerasMNIST

[2]https://github.com/tohinz/SVHN-Classifier

[3]https://github.com/geifmany/cifar-vgg