[Reviews · NeurIPS 2019]

Reviewer 1



Originality: The use of the True Class Probability (TCP) is novel in the area of uncertainty estimation for the task of detecting misclassifications. However, as the TCP cannot be known at test time, the authors use an additional network component to predict this score based intermediate features derived from the predictive model. This is very similar to the task of confidence score estimation in speech recognition (https://arxiv.org/abs/1810.13024, https://arxiv.org/abs/1810.13025) . While the use of the TCP as a target is novel, confidence score estimation is not, thus the work has limited novelty. Quality: The paper is technically sound, experiments are sensible and in line with standard practice in the area. The authors are generally honest about the evaluation of their work, however, they do not analyse the limitations of the given approach. Clarity: The paper is very clearly written, easy to understand and pleasant to read. The author place the work well within the context of current work. To address the authors' question regarding use of information theoretic uncertainty measures for misclassification detection (line 158) - it was shown by Malinin and Gales (https://arxiv.org/abs/1802.10501) that they perform worse than MCP for Misclassification detection. Significance: The proposed method, while sensible and well evaluated, does not seem to provide a significant advantage over baseline approaches as the complexity of the task grows. One of the issues with the proposed approach, which the authors do not discuss, is that the model is trained to predict the TCP on the training data. Thus, if the model is very good and fits the training data well, then the TCP will be, more often than not, equal to the MCP. Thus, there will only be a few cases where the confidence scores predicted by the model will differ significantly from the MCP. Some ways in which this can be remedied is to use some kind of meta-learning style approach and trained the ConfidNet module on holdout data not used to train the predictive model. Additionally, maybe it is possible to balance the loss, such that it is more sensitive to misclassifications than to correct classifications. Another limitation of the proposed approach is that the theoretical guarantees derived for the TCP do not necessarily hold for the confidence scores predicted by the model. Reasons to accept: Very well written paper, pleasant to read. Approach is sensible, good experimental evaluation. Reasons to reject: Limited originality - related to speech recognition confidence score estimation, but such work is not cited. Method does not provide significant gains over baseline approaches. ---POST REBUTTAL COMMENTS--- I will change my rating to a 7 and this this paper should be accepted. I actually quite liked reading this paper and thought that it was professionally written and executed. Furthermore, the authors have engaged with the reviewers concerns and addressed them/provided new results. I still feel that the method lacks a certain degree of novelty and the gains and not as great relative to baseline models. At the same time, I am aware that 'predicting your own mistakes' is a rather challenging task. Furthermore, the experiments are extensive and have been further expanded. Overall, I feel like this paper is a good demonstration of 'good science practice', and thus, I vote accept.

Reviewer 2



I've read the author response and increased my score to a 7 - I vote for acceptance, conditional on the authors including coverage accuracy curves in the final version as they agreed to in the response, and coverage-accuracy numbers (something like table 2 in https://arxiv.org/pdf/1901.09192.pdf would suffice) in the supplementary. I think these are important mainly because past work on selective classification use these, and it would be very helpful for future researchers to compare these numbers. The author response addresses my concerns. It would be good to add intuition about why TCP does better than BCE as a conjecture (future work can check this). The new comparisons against baselines, and TCP on the validation set, were helpful. It would be good to add a note that there could be better ways of using the validation set that maintain accuracy and do even better on your metrics. Suggestions for improvement: - Run on more datasets. BCE is the obvious baseline (train a classifier to predict correct vs wrong examples). The supplementary shows that TCP does better than BCE on 3 datasets, and about 0-2% better. This isn't statistically significant even at p = 0.1. It would be nice if the authors had a chance to include at least one more dataset (even if TCP turns out to be worse than BCE there, that would be good to know), and mention these results in the main paper. - To make the paper even more compelling, I would advocate seeing if this idea still helps when train/test are different domains e.g. MNIST -> SVHN or for out of domain detection. ---------------------------------- Novelty: Medium. Their goal is to identify incorrectly classified examples. The naive baseline is to directly predict whether an example is correctly or incorrectly classified. This is similar to their BCE loss, and has been done before (e.g. Blatz et al 2004, Devries and Taylor 2018). Their specific novelty is to predict the softmax probability of the true label. Also they apply it to in-domain selective classification. The novelty is satisfactory, but not particularly high. Clarity: High. The paper is very clearly written. Quality: Medium. - In practice, neural networks are often trained until they achieve 0 training error. In that case, max-confidence and true-confidence on the training set are identical, because the model gets all the examples correct. Did you stop training your models before they reached 0 training error? My concern is that the method may be less applicable for modern neural networks where we typically train until they get 0 error. At the very least, this caveat should be discussed in the paper, since the method seems sensitive to the training procedure. - It seems like you train the model and the true confidence predictor on the same training set? What if you train the true confidence predictor on the validation set? Could this improve the results, and make it less sensitive to training procedure? My intuition is that this would be better - at least for the naive baseline/BCE where we try and classify an example as being predicted correctly or incorrectly. - What were the test set accuracies of your models? It could potentially be easier to predict incorrect examples for a model that’s less accurate, so it would be good to see these. - The improvement of the proposed method over BCE seems fairly small 1% - 2.5% improvement in AUPR, and BCE seems to be the obvious baseline. While BCE isn’t exactly what DeVries and Taylor did, it’s fairly close. - I think the experiments were fairly extensive. One could always ask for more experiments, for example the method could be compared to DeVries and Taylor, or other selective prediction methods like (Geifman and El-Yaniv, 2017). However, it seems satisfactory to me. It would be nice to see coverage/accuracy curves/scores though (El-Yaniv and Wiener, 2010). - I could not find the GitHub link to the code, even though the reproducibility checklist says it was provided. The appendix says 'all models are trained in a standard way' - more details are needed for reproducibility. Significance: Medium. The problem of identifying incorrectly classified examples, also known as selective prediction, is important. They show that a simple method can work well. The improvements are modest, but the method is a lot simpler than the alternatives so future work may build off of it. References: Confidence Estimation for Machine translation. John Blatz et al. COLING 2004. Learning Confidence for Out-of-Distribution Detection in Neural Networks. DeVries and Taylor. Arxiv 2018. Selective Classification for Deep Neural Networks. Geifman and El-Yaniv. NIPS 2017. On the Foundations of Noise-free Selective Classification. El-Yaniv and Wiener. JMLR 2010.

Reviewer 3



Update after author rebuttal: The authors have addressed concerns over use of held-out set for confidence estimator training and it would definitely be useful if this discussion is added to paper. The authors mention challenges in adopting such an approach with small dataset tasks. However I would recommend a focus on large scale datasets where even preliminary experiments by the author show relatively more interesting observations. 2. Over calibration issues in the confidence estimation branch the authors have reported better calibration performance compared to MCP. However as these details are not clear from the rebuttal, I cannot comment further. I would encourage the readers to not limit their discussion on calibration to a comparison with MCP and provide a discussion on calibration issues observed if any. ------------------------------------------------ In this paper the authors propose a confidence estimation network which is trained for a classifier and shares parameters with the classifier network. The authors are interested in confidence estimation when the model is used matched conditions. Hence they train even the confidence estimator using the same training data. However models typically perform very well on the training data, even when compared to matched test data (i.e., test data drawn from the same true distribution). Hence it is not clear if choosing a held-out set for training the confidence estimator might result in better confidence estimation on test data. It would be very interesting to see if even the confidence estimation network suffers from the same calibration issues as the primary classification neural network. Related analysis would be very informative. Further analyzing the behavior of these confidence estimation models as the test data deviates from the training data would be useful. It is not clear why the confidence estimator needs to share the parameters with the primary classification network. Is it for reducing the computational complexity ?

[Author Response · NeurIPS 2019]

We thank the reviewers for their meaningful and valuable comments, which help to improve the quality of our work.

**Training with few errors [R1, R2, R3]:** Given the small number of errors available to train ConfidNet due to deep
neural network (DNN) over-fitting, one common suggestion from reviewers is to use hold-out data. We performed
preliminary experiments of this variant at submission time and they were not conclusive. We report here a consolidated
evaluation on all datasets to fulfill reviewer's request, shown in the table below for validation sets with 10% of samples.
We observe a general performance drop when using a validation set for training TCP confidence. The drop is especially
pronounced for small datasets (MNIST), where models reach > 97% train and val accuracies. Consequently, with a
high accuracy and a small validation set, we do not get a larger absolute number of errors using val set compared to
train set. One solution would be to increase validation set size but this would damage model's prediction performance.
By contrast, we take care with our approach to base our confidence estimation on models with levels of test predictive
performance that are similar to those of baselines (R2), and on a par with those reported in other papers, *e.g.* Trust
Score (ref. [15] in submission). On CIFAR-100, the gap between train accuracy and val accuracy is substantial (95.56%
vs. 65.96%), which may explain the slight improvement for confidence estimation using val set. We think that training
ConfidNet on val set with models reporting low/middle test accuracies could improve the approach. We would be
glad to add this discussion in the paper if accepted. Note that, in discussed future work, we also consider the use of
adversarial attacks, image corruption or label noise to generate additional errors to train from.

| AUPR-Error (%) | MNIST MLP | MNIST SmallConvNet | SVHN SmallConvNet | CIFAR-10 VGG-16 | CIFAR-100 VGG-16 | CamVid SegNet |
|---|---|---|---|---|---|---|
| ConfidNet (using train set) | 57.34% | 43.94% | 50.72% | 49.94% | 73.68% | 50.28% |
| ConfidNet (using val set) | 33.41% | 34.22% | 47.96% | 48.93% | 73.85% | 50.15% |

**Positioning of the approach [R1, R2]:** We thank R1 and R2 for bringing to our attention related papers on confidence
estimation, we will update references accordingly. R1 mentions the use of bi-directional lattice RNN specifically
designed for confidence estimation in speech recognition, whereas ConfidNet offers a model- and task-agnostic approach
which can be plugged into any DNN. R2: One of the approaches from Blatz *et al.*'04 is similar to our BCE baseline but
is not dedicated to training DNNs. DeVries & Taylor'18 work differs from ours since they perform joint training of
confidence and classification for out-of-distribution detection (l. 166-169 in our paper). In addition, they use predicted
confidence score to interpolate output probabilities and target whereas we specifically defined TCP, a criterion suited for
failure prediction. Finally, post-hoc selective classification methods (R2: Gefman & El-Yaniv'17) identify a threshold
over a confidence-rate function (*e.g.*, MCP) to satisfy a user-specified risk level, whereas we focus on relative metrics.
The approach is compatible with ours and we consider integrating ConfidNet as confidence-rate function in future work.
**Comparison with additional baselines [R2]:** As suggested, we have implemented the approach of DeVries and
Taylor, using their code, for an additional comparison on CIFAR-10 and CIFAR-100. This method obtains resp. 46.07%
and 71.16% on AUPR-Error, similar to other baselines but below ConfidNet (49.94% and 73.68%). This confirms that
their approach is not specifically designed for failure prediction, unlike ours. Results for all datasets will be reported in
Table 1 in the paper. Following R2's suggestions, we will also add coverage-accuracy graphs in supplementary.
**Biasing ConfidNet towards misclassifications [R1]:** We have performed additional experiments for training Confid-
Net with a weighted loss between erroneous and correct predictions, which will be added to supplementary. While
ConfidNet trained with BCE presents small improvements, it does not improve TCP regression. Including an instance-
based weighting scheme using TCP confidence for training would be an interesting direction for future work.
**Improved performances of learning TCP over BCE approach [R2]:** In our setting, ConfidNet is trained to match
TCP criterion thanks to a regression loss. We have the intuition that TCP regularizes training by providing more
fine-grained information about the quality of the classifier regarding a sample's prediction. This is particularly useful in
difficult learning cases where there are only few error samples in training set.
**Reproducibility [R2]:** We will provide link to a GitHub repository with the code and add more implementation details
(hyperparameters, train/val split, accuracies) in supplementary to facilitate reproducibility.
**Effect on calibration [R3]:** Following R3's suggestion, we have studied the effect of our approach on calibration. On
CIFAR-100, it turns out that ConfidNet improves calibration over using MCP as confident estimate (15.61% vs. 22.37%
on ECE). We have obtained similar results for other datasets. We will include these additional results in supplementary.
**Parameters sharing [R3]:** Using a network pre-trained for classification indeed reduces computational complexity
Besides, we observed that it helps ConfidNet learn most specific layers for confidence estimation (l. 108-110). Hence,
this initialization allows a better structuring of the parameter space.
**ConfidNet on mismatch conditions [R3]:** In case of data distribution shift, performance is likely to drop. Since TCP
tends to be less overconfident than MCP on predictions, we expect it to fail more "graciously", though it will eventually
suffer like the main classification branch it is attached to. Leveraging dedicated domain adaptation techniques might
help to overcome the problem, which is an interesting direction for future work.

[Meta-Review · NeurIPS 2019]

The paper proposes a way to learn model confidence, and shows that the proposed method better predicts mis-classifications ("failures"). All the reviewers voted to accept the paper. The reviewers raised a couple of questions initially: the authors have already addressed some of these questions in their rebuttal, which is appreciated. I encourage the authors to take into account the remaining suggestions when preparing the final version.